# *Hunting for Lost Heritage*
## on Wikimedia Commons and Wikidata

*What if we apply data mining on Wikimedia projects to build a more comprehensive open catalogue of Cultural heritage in Italy from crowdsourced contents?*

a personal project - Marco Chemello 2023-25

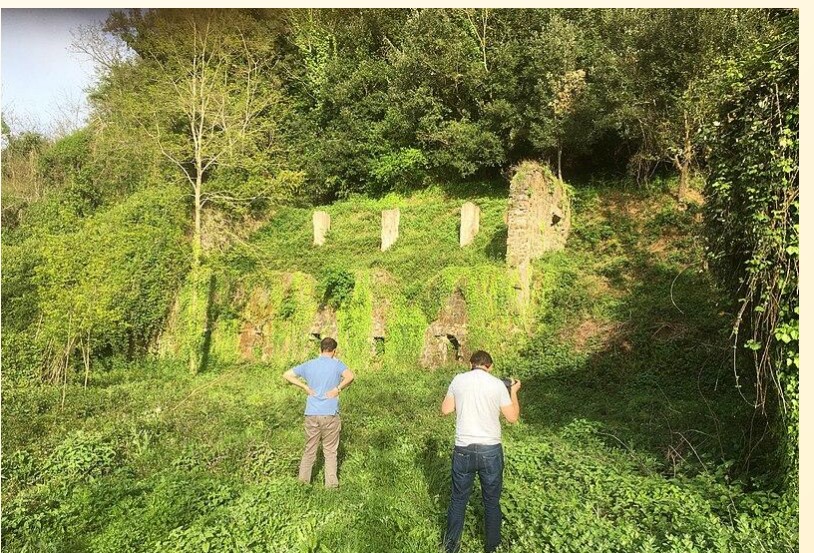

Photo: [Wikigita in Calabria primavera 2022 partecipanti f05](#)
by [Marco Chemello (Wikimedia Italia)](#) 2022
CC BY-SA 4.0 on Wikimedia Commons

# Still a lot of Commons categories not linked to Wikidata...

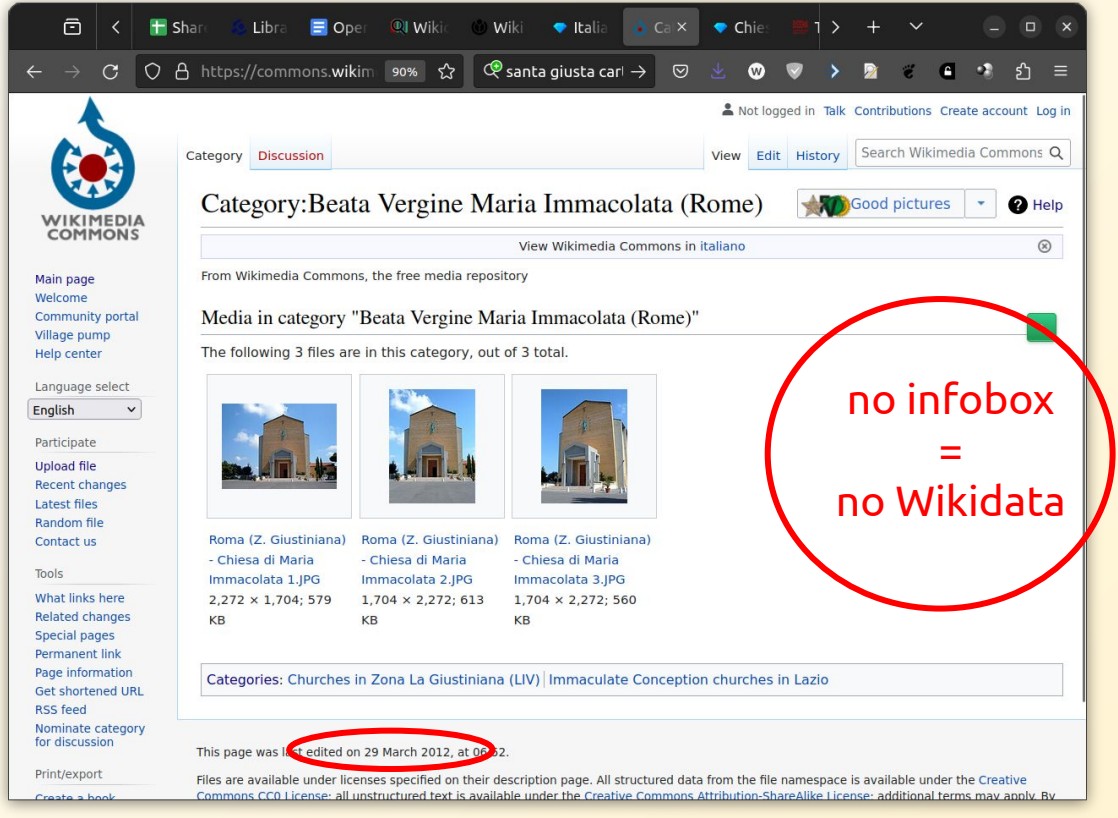

# Still a lot of Commons categories not linked to Wikidata…

Many of the building are really present on Wikidata with an item, so we need to create a link…

… but how many other are not in Wikidata?

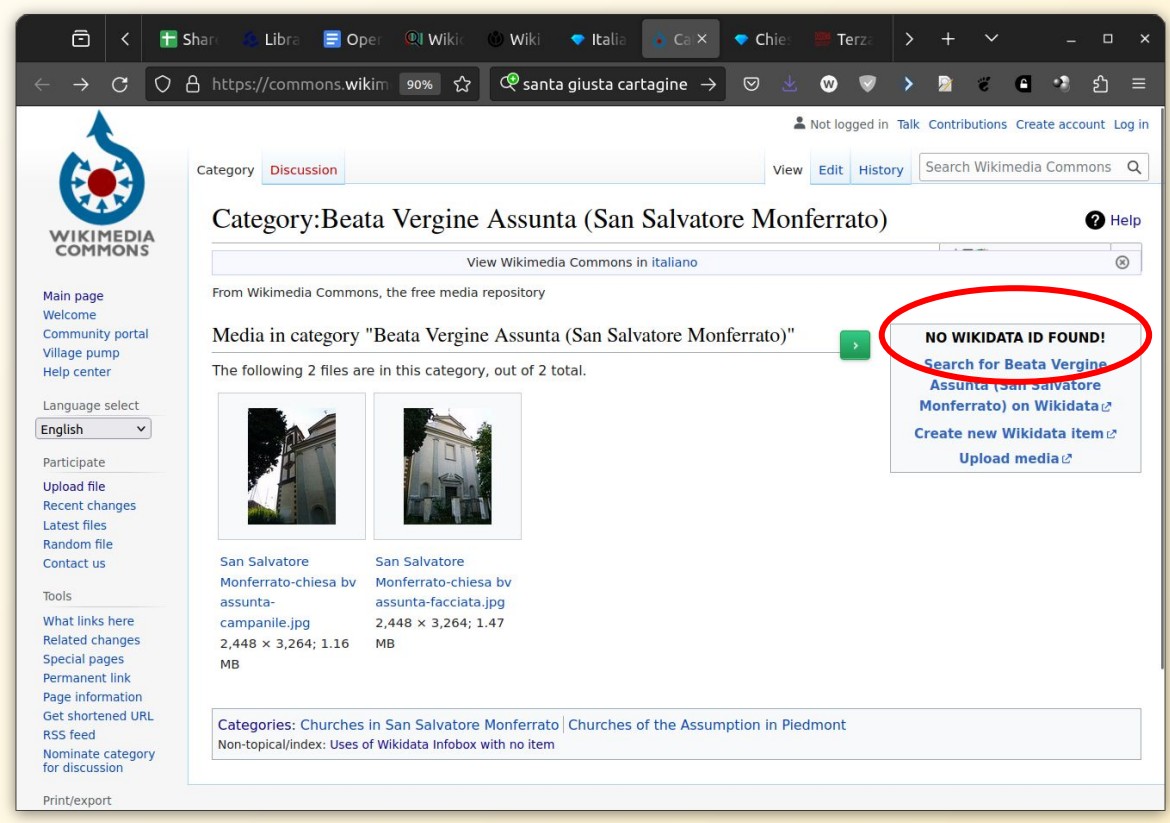

# *Phase 1*
## Hunting for lost churches in Italy

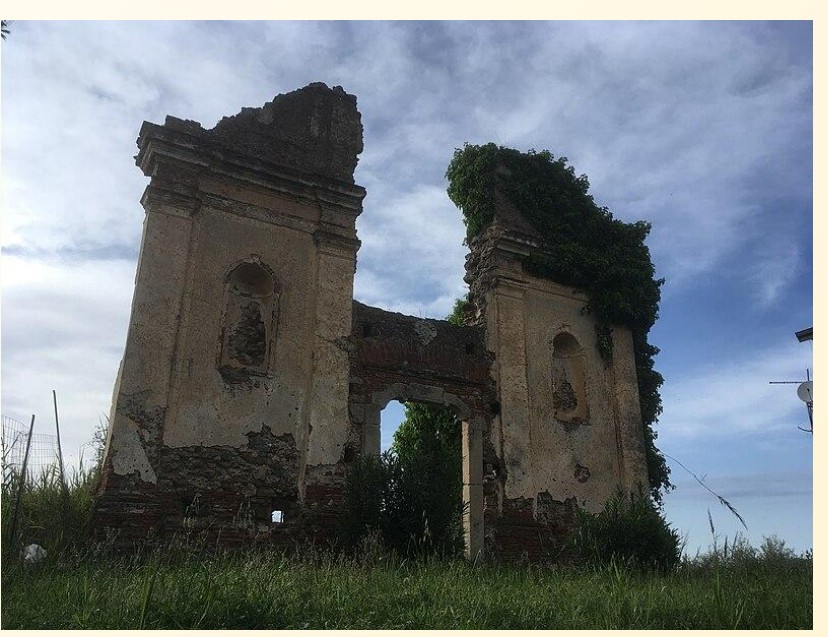

# Wiki Loves Monuments: a lot of church buildings in Italy…

From the results of the 2023 edition of Wiki Loves Monuments Italy:

**64 555** religious buildings in **Italy** present in the WLM lists

The list is based
on **Wikidata**
(now the most comprehensive
database on this topic!)

…but there are still
many gaps to fill!

https://wiki.wikimedia.it/wiki/Wiki_Loves_Monuments/2023

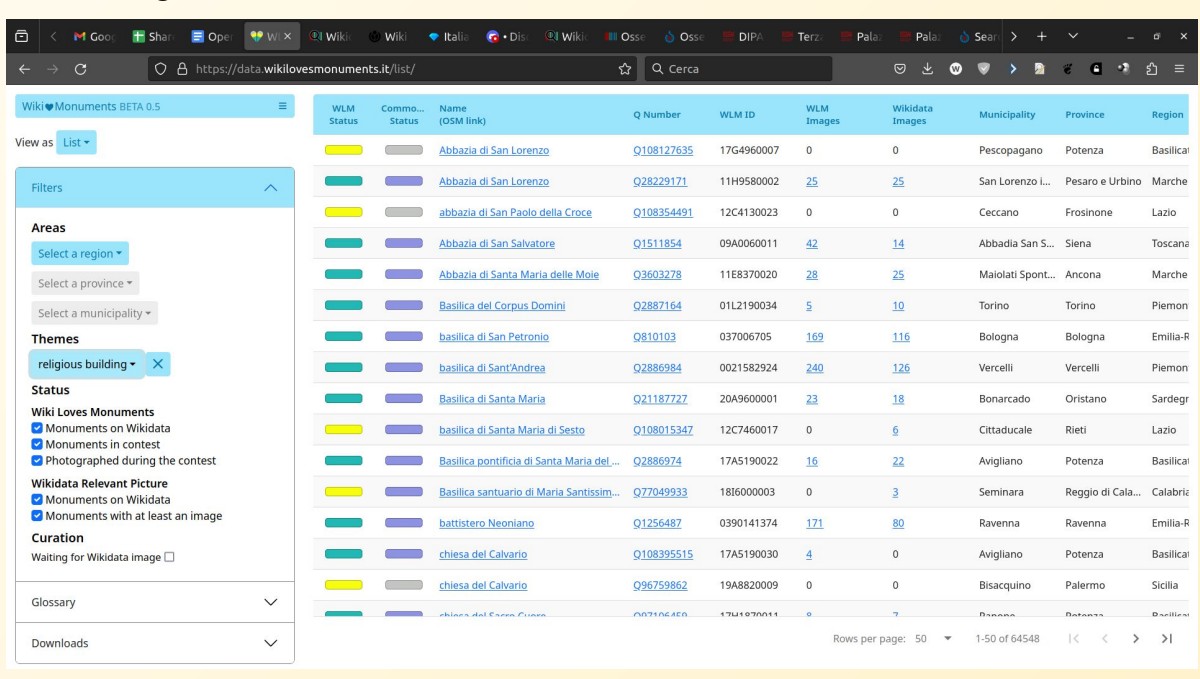

# Methodology

1. Collecting the data from Wikimedia Commons
2. Filtering the data
3. Importing & deriving informations
4. reconciling to Wikidata with existing items
5. creating missing items on Wikidata

# 1. Collecting the data (roughly)
## performing some search on Wikimedia Commons

Do you know better ways to do so?

(yes, there are better ways to try!)

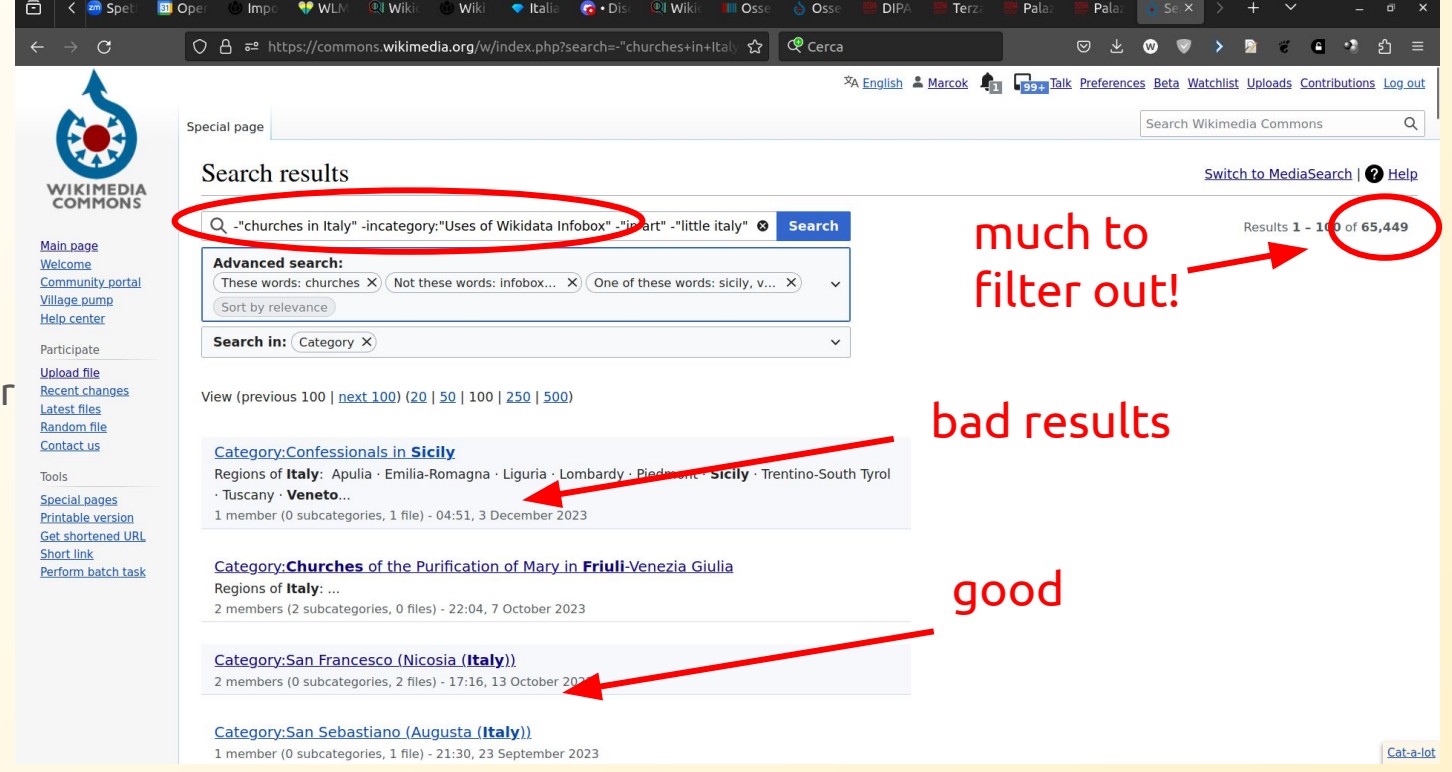

much to filter out!

bad results

good

# 1.  Collecting the data (roughly)
## using PetScan to select categories on Wikimedia Commons

yes, PetScan could
do it better…
but it doesn't work with
big lists,
and/or with too much
filters

better for a
single
region

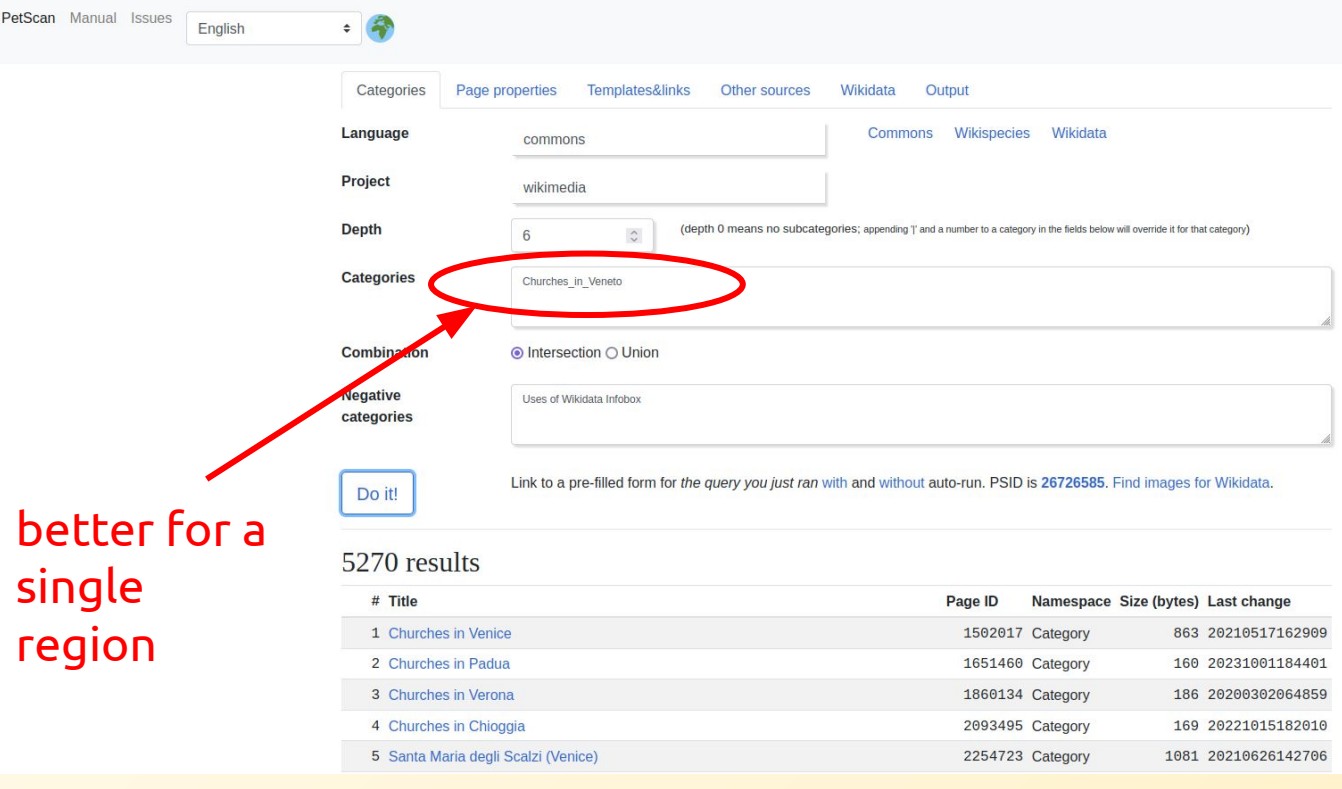

# 2. Filtering the data (roughly)
## pasting, ordering and manually filtering the list of categories

excluding duplicates and categories
of paintings, interiors, architectural elements etc.

Apri ⌄    □        **Chiese 1 A**                    Salva  ☰  —  □  ✕
                   ~/Documenti

 1 Name     info
 2 Abbazia di San Clemente a Casauria - Church
 3 Agonizzanti (Patti)     Saint Mary churches in Sicily
 4 Angelo Raffaele (Venice)
 5 Anime Sante (Alcamo)     Saints Souls of Purgatory churches in Sicily
 6 Annunziata (Airola)     Churches of the Annunciation in Campania
 7 Annunziata (Casalvecchio Siculo)      Churches of the Annunciation in Sicily
 8 Annunziata (Castelmola) Churches of the Annunciation in Sicily
 9 Annunziata (Fiumedinisi)       Churches of the Annunciation in Sicily
10 Annunziata (Meri)     Churches of the Annunciation in Sicily
11 Annunziata (Teramo)     Churches of the Annunciation in Abruzzo
12 Antica Basilica di San Giovanni Battista (Monza)      Churches in Italy built in 595
13 Ascensione (Faveto, Recco)     Churches of the Ascension of Christ in Italy
14 Assunzione di Maria Vergine (Rimagna, Monchio delle Corti)     Baroque churches in Emilia-Romagna
15 Baptistery (Craveggia)  Baptistries in Piedmont
16 Baptistry (Castiglione Olona)   Baptistries in Lombardy
17 Baptistry (Pistoia)     Baptistries in Tuscany
18 Baptistry (Volterra)    Baptistries in Tuscany
19 Basilica di San Pietro Alli Marmi     Saint Peter churches in Campania
20 Beata Vergine Addolorata (Collagù, Farra di Soligo)     Saint Mary churches in Veneto
21 Beata Vergine Assunta (San Salvatore Monferrato)      Churches of the Assumption in Piedmont
22 Beata Vergine delle Grazie (Gemona del Friuli)
23 Beata Vergine del Rosario (Codigoro)    Saint Mary churches in Emilia-Romagna
24 Beata Vergine Maria Addolorata (Racale) Our Lady of Sorrows churches in Apulia
25 Beata Vergine Maria Immacolata (Rome)   Immaculate Conception churches in Lazio
26 Beato Giacomo    Sanctuaries in Apulia
27 Cappella a San Sebastiano (Syracuse)    Saint Sebastian churches in Sicily
28 Cappella dell'Apocalisse (Albanella)    Saint John the Evangelist churches in Campania
29 Cappella delle Rondini (Alviano)     Chapels in Umbria
30 Cappella del Santissimo Salvatore a Pazzano     Former churches in Calabria

                              Testo semplice ⌄  Larg. tab.: 8 ⌄      Rg 15, Col 48    ⌄    INS

# 3. Importing & deriving informations on OpenRefine
## (with a lot of fixing) from category name

➔ Commons category
➔ name (label)
➔ description (English / Italian)
➔ city
➔ location
➔ titular saint

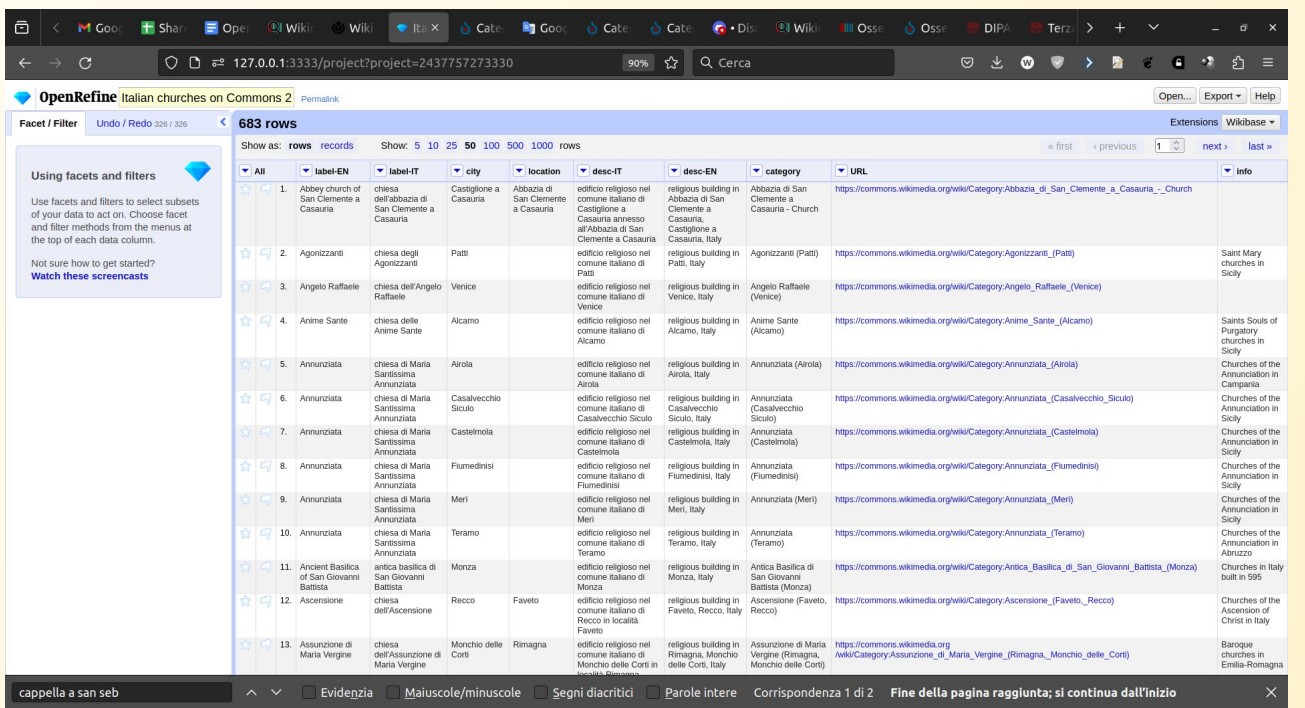

# 4. reconciling to Wikidata with existing items of churches
(tried different ways to match better, as the name variants are a lot)

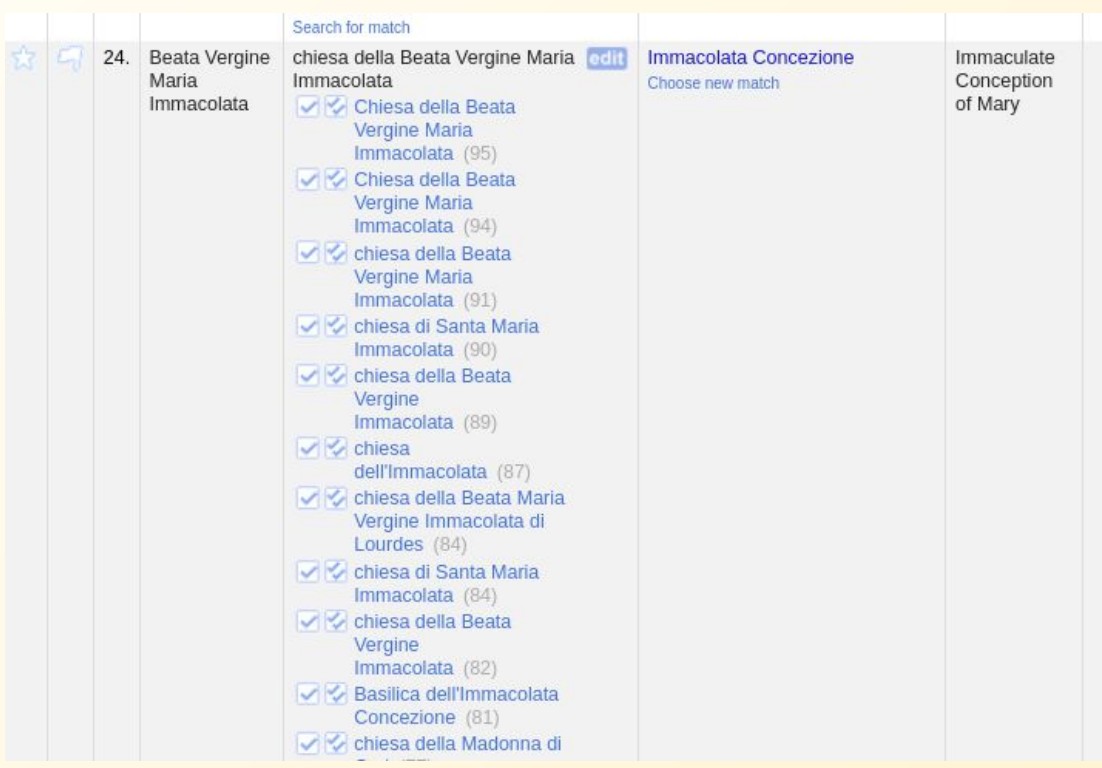

# 5. creating missing items of churches and hamlets
## and updating many more

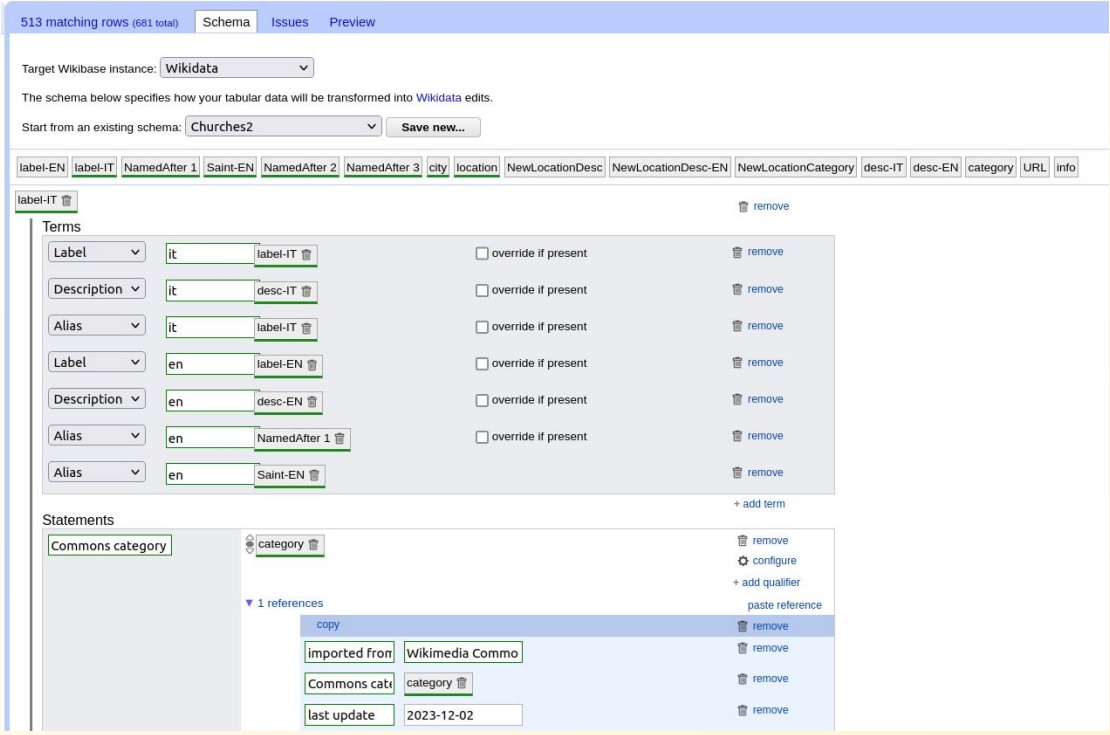

# results of phase 1

(3 months - Oct.-Dec 2023)

**6500+** categories found (with no link in Wikidata)

**3000+ new items** created on Wikidata
(previously unknown religious buildings = +5%)

**updated 4000+** existing items on Wikidata on works
(category link, descriptions, aliases, locations, titular
saints) and artists

(as of 5 Jan. 2024)

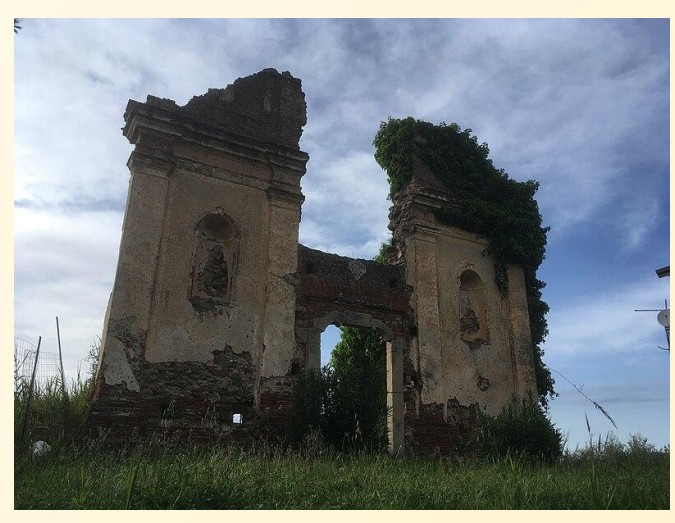

Photo: Rovine Condolima Monterosso
Calabro Wikigita Calabria 2022 f02
by Marco Chemello (Wikimedia Italia) 2022
CC BY-SA 4.0 on Wikimedia Commons

# *Phase 2*

## Hunting for more cultural heritage:
## inside museums and monumental cemeteries

# results of phase 2

(10 months - 8 May 2024 - 15 March 2025)

**10 000+** categories found (with no link in Wikidata)

**9 000+ new items** created on Wikidata (statues, paintings, archaeological artifacts, tombs, exhibits etc.) mainly in Italy

**updated 1 500+** existing items (Commons category, descriptions it/en, aliases, depicts, main subject, commemorates)

(as of 15 March 2025)

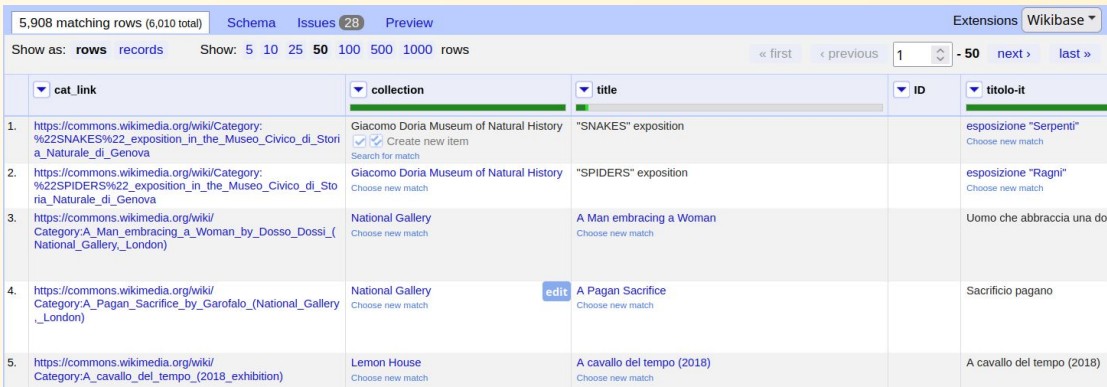

# *Phase 3*
## Using AI to speed-up the process

➔ Performing tests with multiple AI platforms to find best models and prompts

Sei in grado di individuare delle categorie di Wikimedia Commons che riguardano i beni culturali (come musei, monumenti, palazzi, chiese ecc.) che non hanno ancra un collegamento a Wikidata? Ti faccio 2 esempi. 1) Esempio di categoria con collegamento a Wikidata: https://commons.wikimedia.org/wiki/Category:Acquaiolo_(Vincenzo_Gemito) 2) Esempio di categoria senza collegamento a Wikidata: https://commons.wikimedia.org/wiki/Category:Perseus,_Andromeda_and_Cepheus,_Apulian_volute-krater_(Museo_Archeologico_di_Taranto)

Sì, è possibile individuare categorie su Wikimedia Commons relative a beni culturali—come musei, monumenti, palazzi e chiese—che non hanno ancora un collegamento a Wikidata. Queste categorie possono essere identificate attraverso specifiche categorie di manutenzione su Commons che raccolgono le categorie senza link a Wikidata.

**Esempi di categorie senza collegamento a Wikidata:**

1. Category:Perseus, Andromeda and Cepheus, Apulian volute-krater (Museo Archeologico di Taranto)
   Questa categoria, dedicata a un cratere apulo raffigurante Perseo, Andromeda e Cefeo, conservato nel Museo Archeologico di Taranto, non è attualmente collegata a un elemento Wikidata.

**Come individuare altre categorie simili:**

Su Wikimedia Commons esistono categorie di servizio che aiutano a identificare le categorie senza collegamenti a Wikidata. Ad esempio, la Category:Commons category without a link on Wikidata raccoglie tutte le categorie che non hanno un elemento corrispondente su Wikidata. Navigando all'interno di questa categoria, è possibile trovare sottocategorie relative a beni culturali privi di collegamento a Wikidata. Wikipedia Semplice

**Importanza del collegamento a Wikidata:**

Collegare le categorie di Wikimedia Commons a elementi Wikidata è fondamentale per migliorare l'interoperabilità dei dati tra i progetti Wikimedia. Questo permette una migliore organizzazione delle informazioni e facilita l'accesso a dati strutturati, arricchendo l'esperienza degli utenti e contribuendo alla diffusione della conoscenza libera.

**Come contribuire:**

Se individui categorie su Commons relative a beni culturali senza collegamento a Wikidata, puoi contribuire creando un nuovo elemento su Wikidata e aggiungendo il collegamento appropriato. Questo processo aiuta a migliorare la qualità dei dati nei progetti Wikimedia e supporta la comunità nella costruzione di un patrimonio culturale digitale più completo e accessibile.

Fonti Ⓦ

Sei in grado di produrre una lista di categorie con il relativo link? Comincia con 5 categorie sui monumenti e beni culturali.

Certamente! Ecco una lista di 5 categorie su Wikimedia Commons relative a monumenti e beni culturali:

1. Category:Ancient Roman monuments in Italy
   Questa categoria raccoglie immagini e file multimediali relativi ai monumenti dell'antica Roma presenti in Italia.

2. Category:Castles in France
   Include fotografie e media riguardanti i castelli situati in Francia.

3. Category:Historic palaces in Spain
   Contiene file multimediali sui palazzi storici presenti in Spagna.

4. Category:Churches in Germany
   Questa categoria comprende immagini e altri media delle chiese situate in Germania.

5. Category:Museums in Japan
   Raccoglie file multimediali relativi ai musei presenti in Giappone.

Queste categorie offrono una vasta gamma di risorse visive sui monumenti e beni culturali nei rispettivi paesi.

Fonti

Della tua lista, la prima e la terza categoria non esistono su Commons, mentre le altre hanno già il collegamento a Wikidata tramite il template {{Wikidata Infobox}}. Riprova.

🔍 Cercato informazioni su "Category:Palazzo Piovene (Vicenza)"

◎ Sto esplorando le opzioni per migliorare l'efficienza energetica della mia casa, considerato che i costi sono in aumento.

◉ Più letture effettuate da commons.wikimedia.org

🔍 Cercato informazioni su "Category:Museo civico di Eboli"

◎ Sto cercando informazioni sul Museo Archeologico Nazionale di Eboli e media valle del Sele. Mi chiedo quali tesori storici potrebbe racchiudere.

◎ Sto cercando categorie di musei speciali per trovare incongruenze nelle infobox. Sono curioso di esplorare altre opzioni.

🔍 Cercato informazioni su "Category:Monumento ai Caduti" su commons.wikimedia.org

◎ Sto pensando di scegliere uno dei monumenti ai caduti in diverse città per esaminare se abbiano un'infobox.

◉ Più letture effettuate da commons.wikimedia.org

◎ Sto esaminando l'elenco e valutando delle città che verosimilmente non hanno un item associato. Storia e dettagli locali sono in primo piano.

◉ Più letture effettuate da commons.wikimedia.org

◎ Sto cercando di valutare se "Monumento ai Caduti (Corporeno)" o "Spilinga" possano perdere un'infobox per essere piccoli luoghi.

◉ Più letture effettuate da commons.wikimedia.org

◎ Spilinga ha Q126182007, quindi l'oggetto non è smarrito.

Sei in grado di individuare delle categorie di Wikimedia Commons che riguardano i beni culturali (come musei, monumenti, palazzi, chiese ecc.) che non hanno ancra un collegamento a Wikidata? Ti faccio 2 esempi. 1) Esempio di categoria con collegamento a Wikidata: https://commons.wikimedia.org/wiki/Category:Acquaiolo_(Vincenzo_Gemito) 2) Esempio di categoria senza collegamento a Wikidata: https://commons.wikimedia.org/wiki/Category:Perseus,_Andromeda_and_Cepheus,_Apulian_volute-krater_(Museo_Archeologico_di_Taranto)

Thought for 68 seconds ^

Okay, the user is asking me to identify Wikimedia Commons categories related to cultural heritage—like museums, monuments, palaces, churches, etc.—that don't have a link to Wikidata. They provided two examples: one with a Wikidata link and one without. My task is to find more of the latter.

First, I need to understand how Commons categories link to Wikidata. Usually, a category has a "Wikidata item" link in the sidebar. If it's missing, then there's no connection. The user wants categories that lack this link.

I should start by thinking about how to efficiently find such categories. Manually checking each one isn't feasible. Maybe there's a tool or query that can help. I remember that Wikimedia Commons has tools like PetScan and the Commons category tree. Also, maybe SPARQL queries on Wikidata can help find items without a Commons category link, but the user wants the opposite: Commons categories without a Wikidata item.

Wait, maybe I can use the CatScan tool (part of PetScan) on Toolforge. Let me think. If I can search Commons categories under a certain parent category (like "Cultural heritage") and filter those without a Wikidata sitelink. But I'm not sure if CatScan can check for the presence of a Wikidata link. Alternatively, maybe I can use the Wikidata Query Service to find items that are instances of cultural heritage (like museum, monument) but don't have a Commons category link. Wait, but the user wants the inverse: Commons categories without a Wikidata item. That might be harder because Wikidata doesn't track Commons categories directly unless they're linked.

Alternatively, perhaps the user is looking for a way to find Commons categories that should have a corresponding Wikidata item but don't. For example, if a category is about a specific monument, there should be a Wikidata item for that monument, and the category should link to it. If not, then it's an unlinked category.

So, steps to approach this:

1. Identify parent categories on Commons that are related to cultural heritage. For example, "Cultural herit ⌄ ",  "Museums", "Monuments", etc.

⌄

Le categorie che mi hai linkato sopra non esistono, la risposta non è corretta. Prova a trovare 5 categorie esistenti di singoli monumenti italiani /beni culturali su Commons senza collegamento a Wikidata

○ Progetto: **Wikimedia Commons**.
   ○ Categorie genitore: `Monuments in Italy` , `Churches in Italy` , `Villas in Italy` .
   ○ Filtro: **"Has no sitelink to Wikidata"**.
   ○ Depth: 3 (per includere sottocategorie specifiche).

2. **Controllo Manuale**: Ho verificato ogni categoria cliccando sul link e controllando la sidebar di Commons per l'assenza del template Wikidata.

---

### Perché Manca il Collegamento?

Spesso accade per:

● Opere meno note o di contesto locale.
● Categorie create recentemente.
● Errori di mancata sincronizzazione tra progetti.

Se vuoi contribuire, puoi creare gli elementi Wikidata mancanti e collegarli alle categorie! 😊

nessuna delle risposte è corretta.

Thought for 50 seconds ^

Okay, the user is frustrated because my previous attempts to provide correct categories of Italian cultural heritage on Wikimedia Commons without Wikidata links were incorrect. I need to figure out why my previous answers were wrong and how to fix this.

**Ecco cinque categorie di Wikimedia Commons su monumenti e beni culturali italiani prive di collegamento a Wikidata:**

- **Category:Villa Lucertola – Wikimedia Commons** – Categoria dedicata a *Villa Lucertola*, una villa storica a Lezzeno sul Lago di Como, attualmente senza elemento Wikidata collegato COMMONS.WIKIMEDIA.ORG .

- **Category:Palazzo Acrosso Papale – Wikimedia Commons** – Categoria del *Palazzo Acrosso Papale* di Taormina (Sicilia), un palazzo storico privo di collegamento a un item Wikidata COMMONS.WIKIMEDIA.ORG .

- **Category:Monumento ai Caduti (Gallio) – Wikimedia Commons** – Categoria del *Monumento ai Caduti* di Gallio (Vicenza), un monumento commemorativo della Grande Guerra senza alcun collegamento Wikidata COMMONS.WIKIMEDIA.ORG .

- **Category:Museo Diocesano (Tricarico) – Wikimedia Commons** – Categoria riferita al *Museo Diocesano di Tricarico* (Basilicata), museo locale attualmente non connesso ad alcun elemento Wikidata COMMONS.WIKIMEDIA.ORG .

- **Category:Fontana di Nogaredo di Prato – Wikimedia Commons** – Categoria della *fontana di Nogaredo di Prato* (Martignacco, Friuli), un'opera d'arte pubblica non ancora associata ad un item Wikidata COMMONS.WIKIMEDIA.ORG .

Comp⅃ᵢenti! Hai dato tutte risposte corrette.

---

Mi scuso per la confusione precedente. Dopo aver verificato, risulta che la categoria Category:Monumento ai Caduti (Gallio) su Wikimedia Commons non ha un elemento corrispondente su Wikidata. Pertanto, possiamo procedere alla creazione di un nuovo elemento per rappresentare questo monumento.

**Codice QuickStatements per creare l'elemento su Wikidata:**

```sql
                                    Copia    Modifica
CREATE
LAST|Len|"Monumento ai Caduti (Gallio)"
LAST|Lit|"Monumento ai Caduti (Gallio)"
LAST|Den|"war memorial in Gallio, Italy"
LAST|Dit|"monumento ai caduti a Gallio, Italia"
LAST|P31|Q896407
LAST|P17|Q38
LAST|P131|Q55458
LAST|P373|"Monumento ai Caduti (Gallio)"
LAST|P1476|it:"Monumento ai Caduti"
LAST|P625|@45.8667/11.5167
LAST|P143|Q565
LAST|P3452|Q520
LAST|P972|Q4167836
LAST|P813|2025-03-13
```

**Spiegazione del codice:**

# Credits

Presentation made by Marco Chemello in 2023-25

All texts distributed in CC0 1.0 Universal.
All images by the author - distributed with CC BY-SA licenses (see captions).
Trademarks and logos cited for educational purposes.

marco.chemello@wikimedia.it
www.wikimedia.it

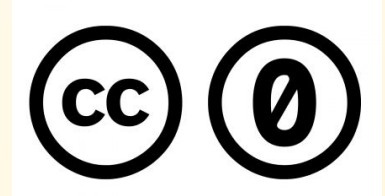

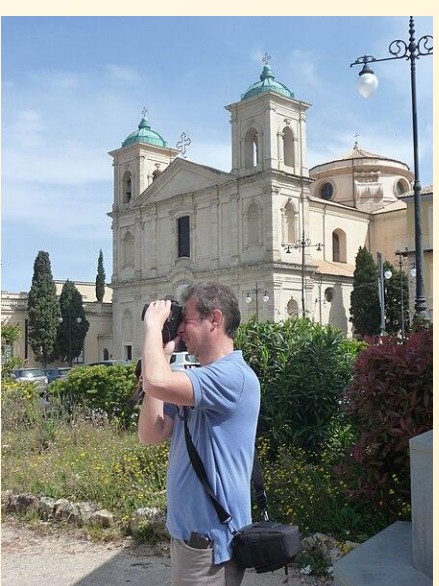

Photo: Wikigita in Calabria primavera 2022 partecipanti f00 by Marco Chemello (Wikimedia Italia) 2022
CC BY-SA 4.0 on Wikimedia Commons

# Marco Chemello
## GLAM specialist and trainer

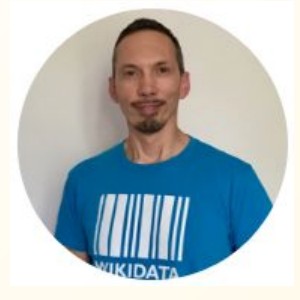

An active contributor to Wikipedia since 2004, he is one of the administrators of the Italian edition of the free encyclopedia since 2005. As a Wikipedian in residence he collaborated on many OpenGLAM projects for sharing the contents of cultural institutions, including the National Museum of Science and Technology Leonardo da Vinci in Milan and the Central Institute for Archives (ICAR) of the Italian Ministry of Culture (MiC). Having joined the Wikimedia Italia staff as a trainer and GLAM specialist, he has been a Wikimedian in residence at the BEIC Foundation, the Academy of Sciences of Turin, the Ricordi Historical Archive of Milan, the Polo del '900. He collaborates with various universities, including the Politecnico and the Statale of Milan. Since 2021 he has been among the first two Wikipedians in residence at an Italian university, the University of Padua. With Unipd he created 3 MOOCs for students, teachers and cultural institutions on EduOpen. In 2024 he's migrating the Museo Egizio di Torino catalogues to Wikimedia projects.