# OpenReview forum: "Hunting for Lost Heritage on Wikimedia Commons and Wikidata"
_wikimedia.it/Wikidata_and_Research/2025/Conference — WD&R Paper_

### Official Review · ~Lucia_Sardo1 · 2025-01-07
**revisione**

**Originality:** 4
**Impact:** 5
**Confidence:** 5

**Review:**

La relazione presenta una proposta operativa di utilizzo di tecniche di data mining per arricchire Wikidata con dati già presenti in Wikimedia Commons. Pur non avendo aspetti fortemente innovativi rispetto alle tematiche trattate, si tratta di una proposta interessante per l'aggiunta di dati a Wikidata e per la valorizzazione di "beni culturali" al momento sotto rappresentati e poco conosciuti. L'autore è sicuramente un esperto del trattamento di queste tipologie di dati in Wikidata e il progetto di cui intende parlare è già avviato, motivo per cui si ritiene valida la proposta e di sicuro interesse, data anche la sua applicabilità ad altre tipologie di dati.

**Compliance:**

5

**Scientific Quality:**

4

---

### Official Review · ~Carlo_Bianchini1 · 2025-01-08

**Originality:** 4
**Impact:** 4
**Confidence:** 4

**Review:**

La presentazione descrive un progetto in tre fasi, delle quali la prima è stata realizzata e viene illustrata. Il lavoro è originale rispetto alla tipologia dei dati che prende in considerazione e dal punto di vista metodologico si avvale di tecniche e strumenti ben conosciuti, mettendo in evidenza le criticità nel loro utilizzo nel caso di studio. Sarebbe interessante che la presentazione fosse integrata con una breve descrizione relativa alla metodologia che il progetto prevede di utilizzare nelle fasi 2 e 3.

**Compliance:**

5

**Scientific Quality:**

4

---

### Decision · Program_Chairs · 2025-02-05

Accept (Paper)